# Distributed $k$-Means and $k$-Median Clustering on General Topologies

**Maria Florina Balcan, Steven Ehrlich, Yingyu Liang**
School of Computer Science
Georgia Institute of Technology
Atlanta, GA 30332
{ninamf,sehrlich}@cc.gatech.edu,yliang39@gatech.edu

## Abstract

This paper provides new algorithms for distributed clustering for two popular center-based objectives, $k$-median and $k$-means. These algorithms have provable guarantees and improve communication complexity over existing approaches. Following a classic approach in clustering by [13], we reduce the problem of finding a clustering with low cost to the problem of finding a coreset of small size. We provide a distributed method for constructing a global coreset which improves over the previous methods by reducing the communication complexity, and which works over general communication topologies. Experimental results on large scale data sets show that this approach outperforms other coreset-based distributed clustering algorithms.

## 1 Introduction

Most classic clustering algorithms are designed for the centralized setting, but in recent years data has become distributed over different locations, such as distributed databases [21, 5], images and videos over networks [20], surveillance [11] and sensor networks [4, 12]. In many of these applications the data is inherently distributed because, as in sensor networks, it is collected at different sites. As a consequence it has become crucial to develop clustering algorithms which are effective in the distributed setting.

Several algorithms for distributed clustering have been proposed and empirically tested. Some of these algorithms [10, 22, 7] are direct adaptations of centralized algorithms which rely on statistics that are easy to compute in a distributed manner. Other algorithms [14, 17] generate summaries of local data and transmit them to a central coordinator which then performs the clustering algorithm. No theoretical guarantees are provided for the clustering quality in these algorithms, and they do not try to minimize the communication cost. Additionally, most of these algorithms assume that the distributed nodes can communicate with all other sites or that there is a central coordinator that communicates with all other sites.

In this paper, we study the problem of distributed clustering where the data is distributed across nodes whose communication is restricted to the edges of an arbitrary graph. We provide algorithms with small communication cost and provable guarantees on the clustering quality. Our technique for reducing communication in general graphs is based on the construction of a small set of points which act as a proxy for the entire data set.

An $\epsilon$-*coreset* is a weighted set of points whose cost on any set of centers is approximately the cost of the original data on those same centers up to accuracy $\epsilon$. Thus an approximate solution for the coreset is also an approximate solution for the original data. Coresets have previously been studied in the centralized setting ([13, 8]) but have also recently been used for distributed clustering as in [23] and as implied by [9]. In this work, we propose a distributed algorithm for $k$-means and $k$-

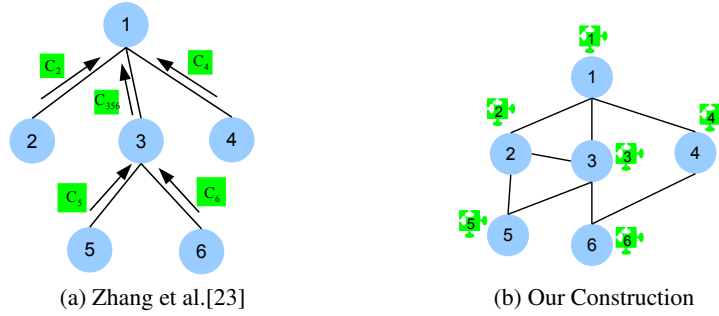

(a) Zhang et al.[23]                    (b) Our Construction

Figure 1: **(a)** Each node computes a coreset on the weighted pointset for its own data and its subtrees' coresets. **(b)** Local constant approximation solutions are computed, and the costs of these solutions are used to coordinate the construction of a local portion on each node.

median, by which each node constructs a local portion of a global coreset. Communicating the approximate cost of a global solution to each node is enough for the local construction, leading to low communication cost overall. The nodes then share the local portions of the coreset, which can be done efficiently in general graphs using a message passing approach.

More precisely, in Section 3, we propose a distributed coreset construction algorithm based on local approximate solutions. Each node computes an approximate solution for its local data, and then constructs the local portion of a coreset using only its local data and the total cost of each node's approximation. For $\epsilon$ constant, this builds a coreset of size $\tilde{O}(kd+nk)$ for $k$-median and $k$-means when the data lies in $d$ dimensions and is distributed over $n$ sites. If there is a central coordinator among the $n$ sites, then clustering can be performed on the coordinator by collecting the local portions of the coreset with a communication cost equal to the coreset size $\tilde{O}(kd + nk)$. For distributed clustering over general connected topologies, we propose an algorithm based on the distributed coreset construction and a message-passing approach, whose communication cost improves over previous coreset-based algorithms. We provide a detailed comparison below.

Experimental results on large scale data sets show that our algorithm performs well in practice. For a fixed amount of communication, our algorithm outperforms other coreset construction algorithms.

**Comparison to Other Coreset Algorithms:** Since coresets summarize local information they are a natural tool to use when trying to reduce communication complexity. If each node constructs an $\epsilon$-coreset on its local data, then the union of these coresets is clearly an $\epsilon$-coreset for the entire data set. Unfortunately the size of the coreset in this approach increases greatly with the number of nodes.

Another approach is the one presented in [23]. Its main idea is to approximate the union of local coresets with another coreset. They assume nodes communicate over a rooted tree, with each node passing its coreset to its parent. Because the approximation factor of the constructed coreset depends on the quality of its component coresets, the accuracy a coreset needs (and thus the overall communication complexity) scales with the height of this tree. Although it is possible to find a spanning tree in any communication network, when the graph has large diameter every tree has large height. In particular many natural networks such as grid networks have a large diameter ($\Omega(\sqrt{n})$ for grids) which greatly increases the size of the local coresets. We show that it is possible to construct a global coreset with low communication overhead. This is done by distributing the coreset construction procedure rather than combining local coresets. The communication needed to construct this coreset is negligible – just a single value from each data set representing the approximate cost of their local optimal clustering. Since the sampled global $\epsilon$-coreset is the same size as any local $\epsilon$-coreset, this leads to an improvement of the communication cost over the other approaches. See Figure 1 for an illustration. The constructed coreset is smaller by a factor of $n$ in general graphs, and is independent of the communication topology. This method excels in sparse networks with large diameters, where the previous approach in [23] requires coresets that are quadratic in the size of the diameter for $k$-median and quartic for $k$-means; see Section 4 for details. [9] also merge coresets using coreset construction, but they do so in a model of parallel computation and ignore communication costs.

Balcan et al. [3] and Daume et al. [6] consider communication complexity questions arising when doing classification in distributed settings. In concurrent and independent work, Kannan and Vem-

pala [15] study several optimization problems in distributed settings, including $k$-means clustering under an interesting separability assumption.

## 2  Preliminaries

Let $d(p,q)$ denote the Euclidean distance between any two points $p, q \in \mathbf{R}^d$. The goal of $k$-means clustering is to find a set of $k$ centers $\mathbf{x} = \{x_1, x_2, \ldots, x_k\}$ which minimize the $k$-means cost of data set $P \subseteq \mathbf{R}^d$. Here the $k$-means cost is defined as $\mathrm{cost}(P, \mathbf{x}) = \sum_{p \in P} d(p, \mathbf{x})^2$ where $d(p, \mathbf{x}) = \min_{x \in \mathbf{x}} d(p, x)$. If $P$ is a weighted data set with a weighting function $w$, then the $k$-means cost is defined as $\sum_{p \in P} w(p) d(p, \mathbf{x})^2$. Similarly, the $k$-median cost is defined as $\sum_{p \in P} d(p, \mathbf{x})$. Both $k$-means and $k$-median cost functions are known to be **NP**-hard to minimize (see for example [2]). For both objectives, there exist several readily available polynomial-time algorithms that achieve constant approximation solutions (see for example [16, 18]).

In distributed clustering, we consider a set of $n$ nodes $V = \{v_i, 1 \le i \le n\}$ which communicate on an undirected connected graph $G = (V, E)$ with $m = |E|$ edges. More precisely, an edge $(v_i, v_j) \in E$ indicates that $v_i$ and $v_j$ can communicate with each other. Here we measure the communication cost in number of points transmitted, and assume for simplicity that there is no latency in the communication. On each node $v_i$, there is a local data set $P_i$, and the global data set is $P = \bigcup_{i=1}^{n} P_i$. The goal is to find a set of $k$ centers $\mathbf{x}$ which optimize $\mathrm{cost}(P, \mathbf{x})$ while keeping the computation efficient and the communication cost as low as possible. Our focus is to reduce the communication cost while preserving theoretical guarantees for approximating clustering cost.

**Coresets:** For the distributed clustering task, a natural approach to avoid broadcasting raw data is to generate a local summary of the relevant information. If each site computes a summary for their own data set and then communicates this to a central coordinator, a solution can be computed from a much smaller amount of data, drastically reducing the communication.

In the centralized setting, the idea of summarization with respect to the clustering task is captured by the concept of coresets [13, 8]. A coreset is a set of weighted points whose cost approximates the cost of the original data for any set of $k$ centers. The formal definition of coresets is:

**Definition 1** (**coreset**). *An $\epsilon$-coreset for a set of points $P$ with respect to a center-based cost function is a set of points $S$ and a set of weights $w : S \to \mathbf{R}$ such that for any set of centers $\mathbf{x}$, we have $(1 - \epsilon)\mathrm{cost}(P, \mathbf{x}) \le \sum_{p \in S} w(p)\mathrm{cost}(p, \mathbf{x}) \le (1 + \epsilon)\mathrm{cost}(P, \mathbf{x}).$*

In the centralized setting, many coreset construction algorithms have been proposed for $k$-median, $k$-means and some other cost functions. For example, for points in $\mathbf{R}^d$, algorithms in [8] construct coresets of size $\tilde{O}(kd/\epsilon^4)$ for $k$-means and coresets of size $\tilde{O}(kd/\epsilon^2)$ for $k$-median. In the distributed setting, it is natural to ask whether there exists an algorithm that constructs a small coreset for the entire point set but still has low communication cost. Note that the union of coresets for multiple data sets is a coreset for the union of the data sets. The immediate construction of combining the local coresets from each node would produce a global coreset whose size was larger by a factor of $n$, greatly increasing the communication complexity. We present a distributed algorithm which constructs a global coreset the same size as the centralized construction and only needs a single value[1] communicated to each node. This serves as the basis for our distributed clustering algorithm.

## 3  Distributed Coreset Construction

Here we design a distributed coreset construction algorithm for $k$-means and $k$-median. The underlying technique can be extended to other additive clustering objectives such as $k$-line median.

To gain some intuition on the distributed coreset construction algorithm, we briefly review the construction algorithm in [8] in the centralized setting. The coreset is constructed by computing a constant approximation solution for the entire data set, and then sampling points proportional to their contributions to the cost of this solution. Intuitively, the points close to the nearest centers can be approximately represented by the centers while points far away cannot be well represented. Thus, points should be sampled with probability proportional to their contributions to the cost. Directly adapting the algorithm to the distributed setting would require computing a constant approximation

**Algorithm 1** Communication aware distributed coreset construction

---

**Input:** Local datasets $\{P_i, 1 \leq i \leq n\}$, parameter $t$ (number of points to be sampled).

    **Round 1:** on each node $v_i \in V$

        • Compute a constant approximation $B_i$ for $P_i$.
          Communicate $\mathrm{cost}(P_i, B_i)$ to all other nodes.

    **Round 2:** on each node $v_i \in V$

        • Set $t_i = \frac{t\,\mathrm{cost}(P_i, B_i)}{\sum_{j=1}^{n} \mathrm{cost}(P_j, B_j)}$ and $m_p = \mathrm{cost}(p, B_i), \forall p \in P_i$.

        • Pick a non-uniform random sample $S_i$ of $t_i$ points from $P_i$,
          where for every $q \in S_i$ and $p \in P_i$, we have $q = p$ with probability $m_p / \sum_{z \in P_i} m_z$.
          Let $w_q = \frac{\sum_i \sum_{z \in P_i} m_z}{t m_q}$ for each $q \in S_i$.

        • For $\forall b \in B_i$, let $P_b = \{p \in P_i : d(p, b) = d(p, B_i)\}$, $w_b = |P_b| - \sum_{q \in P_b \cap S} w_q$.

**Output:** Distributed coreset: points $S_i \cup B_i$ with weights $\{w_q : q \in S_i \cup B_i\}, 1 \leq i \leq n$.

---

solution for the entire data set. We show that a global coreset can be constructed in a distributed fashion by estimating the weight of the entire data set with the sum of local approximations. With this approach, it suffices for nodes to communicate the total costs of their local solutions.

**Theorem 1.** *For distributed $k$-means and $k$-median clustering on a graph, there exists an algorithm such that with probability at least $1 - \delta$, the union of its output on all nodes is an $\epsilon$-coreset for $P = \bigcup_{i=1}^{n} P_i$. The size of the coreset is $O(\frac{1}{\epsilon^4}(kd + \log \frac{1}{\delta}) + nk \log \frac{nk}{\delta})$ for $k$-means, and $O(\frac{1}{\epsilon^2}(kd + \log \frac{1}{\delta}) + nk)$ for $k$-median. The total communication cost is $O(mn)$.*

As described below, the distributed coreset construction can be achieved by using Algorithm 1 with appropriate $t$, namely $O(\frac{1}{\epsilon^4}(kd + \log \frac{1}{\delta}) + nk \log \frac{nk}{\delta})$ for $k$-means and $O(\frac{1}{\epsilon^2}(kd + \log \frac{1}{\delta}))$ for $k$-median. Due to space limitation, we describe a proof sketch highlighting the intuition and provide the details in the supplementary material.

**Proof Sketch of Theorem 1:** The analysis relies on the definition of the pseudo-dimension of a function space and a sampling lemma.

**Definition 2** ([19, 8]). *Let $F$ be a finite set of functions from a set $P$ to $\mathbf{R}_{\geq 0}$. For $f \in F$, let $B(f, r) = \{p : f(p) \leq r\}$. The dimension of the function space $\dim(F, P)$ is the smallest integer $d$ such that for any $G \subseteq P$, $\left|\{G \cap B(f, r) : f \in F, r \geq 0\}\right| \leq |G|^d$.*

Suppose we draw a sample $S$ according to $\{m_p : p \in P\}$, namely for each $q \in S$ and $p \in P$, $q = p$ with probability $\frac{m_p}{\sum_{z \in P} m_z}$. Set the weights of the points as $w_p = \frac{\sum_{z \in P} m_z}{m_p |S|}$ for $p \in P$. Then for any $f \in F$, the expectation of the weighted cost of $S$ equals the cost of the original data $P$, since

$$\mathbf{E}\left[\sum_{q \in S} w_q f(q)\right] = \sum_{q \in S} \mathbf{E}[w_q f(q)] = \sum_{q \in S} \sum_{p \in P} \Pr[q = p] w_p f(p) = \sum_{p \in P} f(p).$$

If the sample size is large enough, then we also have concentration for any $f \in F$. The lemma is implicit in [8] and we include the proof in the supplementary material.

**Lemma 1.** *Fix a set $F$ of functions $f : P \to \mathbf{R}_{\geq 0}$. Let $S$ be a sample drawn i.i.d. from $P$ according to $\{m_p \in \mathbf{R}_{\geq 0} : p \in P\}$: for each $q \in S$ and $p \in P$, $q = p$ with probability $\frac{m_p}{\sum_{z \in P} m_z}$. Let $w_p = \frac{\sum_{z \in P} m_z}{m_p |S|}$ for $p \in P$. For a sufficiently large $c$, if $|S| \geq \frac{c}{\epsilon^2}\left(\dim(F, P) + \log \frac{1}{\delta}\right)$, then with probability at least $1 - \delta, \forall f \in F : \left|\sum_{p \in P} f(p) - \sum_{q \in S} w_q f(q)\right| \leq \epsilon \left(\sum_{p \in P} m_p\right)\left(\max_{p \in P} \frac{f(p)}{m_p}\right).$*

To get a small bound on the difference between $\sum_{p \in P} f(p)$ and $\sum_{q \in S} w_q f(q)$, we need to choose $m_p$ such that $\sum_{p \in P} m_p$ is small and $\max_{p \in P} \frac{f(p)}{m_p}$ is bounded. More precisely, if we choose $m_p = \max_{f \in F} f(p)$, then the difference is bounded by $\epsilon \sum_{p \in P} m_p$.

We first consider the centralized setting and review how [8] applied the lemma to construct a coreset for $k$-median as in Definition 1. A natural approach is to apply this lemma directly to the cost $f_{\mathbf{x}}(p) := \mathrm{cost}(p, \mathbf{x})$. The problem is that a suitable upper bound $m_p$ is not available for $\mathrm{cost}(p, \mathbf{x})$. However, we can still apply the lemma to a different set of functions defined as follows. Let $b_p$ denote the closest center to $p$ in the approximation solution. Aiming to approximate

the error $\sum_p [\text{cost}(p, \mathbf{x}) - \text{cost}(b_p, \mathbf{x})]$ rather than to approximate $\sum_p \text{cost}(p, \mathbf{x})$ directly, we define $f_{\mathbf{x}}(p) := \text{cost}(p, \mathbf{x}) - \text{cost}(b_p, \mathbf{x}) + \text{cost}(p, b_p)$, where $\text{cost}(p, b_p)$ is added so that $f_{\mathbf{x}}(p) \geq 0$. Since $0 \leq f_{\mathbf{x}}(p) \leq 2\text{cost}(p, b_p)$, we can apply the lemma with $m_p = 2\text{cost}(p, b_p)$. It bounds the difference $|\sum_{p \in P} f_{\mathbf{x}}(p) - \sum_{q \in S} w_q f_{\mathbf{x}}(q)|$ by $2\epsilon \sum_{p \in P} \text{cost}(p, b_p)$, so we have an $O(\epsilon)$-approximation.

Note that $\sum_{p \in P} f_{\mathbf{x}}(p) - \sum_{q \in S} w_q f_{\mathbf{x}}(q)$ does not equal $\sum_{p \in P} \text{cost}(p, \mathbf{x}) - \sum_{q \in S} w_q \text{cost}(q, \mathbf{x})$. However, it equals the difference between $\sum_{p \in P} \text{cost}(p, \mathbf{x})$ and a weighted cost of the sampled points and the centers in the approximation solution. To get a coreset as in Definition 1, we need to add the centers of the approximation solution with specific weights to the coreset. Then when the sample is sufficiently large, the union of the sampled points and the centers is an $\epsilon$-coreset.

Our key contribution in this paper is to show that in the distributed setting, it suffices to choose $b_p$ from the local approximation solution for the local dataset containing $p$, rather than from an approximation solution for the global dataset. Furthermore, the sampling and the weighting of the coreset points can be done in a local manner. In the following, we provide a formal verification of our discussion above. We have the following lemma for $k$-median with $F = \{f_{\mathbf{x}} : f_{\mathbf{x}}(p) = d(p, \mathbf{x}) - d(b_p, \mathbf{x}) + d(p, b_p), \mathbf{x} \in (\mathbf{R}^d)^k\}$.

**Lemma 2.** *For $k$-median, the output of Algorithm 1 is an $\epsilon$-coreset with probability at least $1 - \delta$, if $t \geq \frac{c}{\epsilon^2} \left( \dim(F, P) + \log \frac{1}{\delta} \right)$ for a sufficiently large constant $c$.*

**Proof Sketch of Lemma 2:** We want to show that for any set of centers $\mathbf{x}$ the true cost for using these centers is well approximated by the cost on the weighted coreset. Note that our coreset has two types of points: sampled points $q \in S = \cup_{i=1}^n S_i$ with weight $w_q := \frac{\sum_{z \in P} m_z}{m_q |S|}$ and local solution centers $b \in B = \cup_{i=1}^n B_i$ with weight $w_b := |P_b| - \sum_{q \in S \cap P_b} w_q$. We use $b_p$ to represent the nearest center to $p$ in the local approximation solution. We use $P_b$ to represent the set of points which have $b$ as their closest center in the local approximation solution.

As mentioned above, we construct $f_{\mathbf{x}}(p)$ to be the difference between the cost of $p$ and the cost of $b_p$ so that Lemma 1 can be applied. Note that the centers are weighted such that $\sum_{b \in B} w_b d(b, \mathbf{x}) = \sum_{b \in B} |P_b| d(b, \mathbf{x}) - \sum_{b \in B} \sum_{q \in S \cap P_b} w_q d(b, \mathbf{x}) = \sum_{p \in P} d(b_p, \mathbf{x}) - \sum_{q \in S} w_q d(b_q, \mathbf{x})$. Taken together with the fact that $\sum_{p \in P} m_p = \sum_{q \in S} w_q m_q$, we can show that $\left| \sum_{p \in P} d(p, \mathbf{x}) - \sum_{q \in S \cup B} w_q d(q, \mathbf{x}) \right| = \left| \sum_{p \in P} f_{\mathbf{x}}(p) - \sum_{q \in S} w_q f_{\mathbf{x}}(q) \right|$. Note that $0 \leq f_{\mathbf{x}}(p) \leq 2d(p, b_p)$ by triangle inequality, and $S$ is sufficiently large and chosen according to weights $m_p = d(p, b_p)$, so the conditions of Lemma 1 are met. Thus we can conclude that $\left| \sum_{p \in P} d(p, \mathbf{x}) - \sum_{q \in S \cup B} w_q d(q, \mathbf{x}) \right| \leq O(\epsilon) \sum_{p \in P} d(p, \mathbf{x})$, as desired.

In [8] it is shown that $\dim(F, P) = O(kd)$. Therefore, by Lemma 2, when $|S| \geq O\left(\frac{1}{\epsilon^2}(kd + \log \frac{1}{\delta})\right)$, the weighted cost of $S \cup B$ approximates the $k$-median cost of $P$ for any set of centers, then $(S \cup B, w)$ becomes an $\epsilon$-coreset for $P$. The total communication cost is bounded by $O(mn)$, since even in the most general case that every node only knows its neighbors, we can broadcast the local costs with $O(mn)$ communication (see Algorithm 3).

**Proof Sketch for $k$-means:** Similar methods prove that for $k$-means when $t = O(\frac{1}{\epsilon^4}(kd + \log \frac{1}{\delta}) + nk \log \frac{nk}{\delta}))$, the algorithm constructs an $\epsilon$-coreset with probability at least $1 - \delta$. The key difference is that triangle inequality does not apply directly to the $k$-means cost, and so the error $|\text{cost}(p, \mathbf{x}) - \text{cost}(b_p, \mathbf{x})|$ and thus $f_{\mathbf{x}}(p)$ are not bounded. The main change to the analysis is that we divide the points into two categories: good points whose costs approximately satisfy the triangle inequality (up to a factor of $1/\epsilon$) and bad points. The good points for a fixed set of centers $\mathbf{x}$ are defined as $G(\mathbf{x}) = \{p \in P : |\text{cost}(p, \mathbf{x}) - \text{cost}(b_p, \mathbf{x})| \leq \Delta_p\}$ where the upper bound is $\Delta_p = \frac{\text{cost}(p, b_p)}{\epsilon}$, and the analysis follows as in Lemma 2. For bad points we can show that the difference in cost must still be small, namely $O(\epsilon \min\{\text{cost}(p, \mathbf{x}), \text{cost}(b_p, \mathbf{x})\})$.

More formally, let $f_{\mathbf{x}}(p) = \text{cost}(p, \mathbf{x}) - \text{cost}(b_p, \mathbf{x}) + \Delta_p$, and let $g_{\mathbf{x}}(p)$ be $f_{\mathbf{x}}(p)$ if $p \in G(\mathbf{x})$ and 0 otherwise. Then $\sum_{p \in P} \text{cost}(p, \mathbf{x}) - \sum_{q \in S \cup B} w_q \text{cost}(q, \mathbf{x})$ is decomposed into three terms:

$$\underbrace{\sum_{p \in P} g_{\mathbf{x}}(p) - \sum_{q \in S} w_q g_{\mathbf{x}}(q)}_{(A)} + \underbrace{\sum_{p \in P \setminus G(\mathbf{x})} f_{\mathbf{x}}(p)}_{(B)} - \underbrace{\sum_{q \in S \setminus G(\mathbf{x})} w_q f_{\mathbf{x}}(q)}_{(C)}$$

---
**Algorithm 2** Distributed clustering on a graph

---
**Input:** $\{P_i, 1 \leq i \leq n\}$: local datasets; $\{N_i, 1 \leq i \leq n\}$: the neighbors of $v_i$; $\mathcal{A}_\alpha$: an $\alpha$-approximation algorithm for weighted clustering instances.
      **Round 1:** on each node $v_i$
            &bull; Construct its local portion $D_i$ of an $\epsilon/2$-coreset by Algorithm 1,
              using Message-Passing for communicating the local costs.
      **Round 2:** on each node $v_i$
            &bull; Call Message-Passing($D_i, N_i$). Compute $\mathbf{x} = \mathcal{A}_\alpha(\bigcup_j D_j)$.
**Output:** $\mathbf{x}$

---

---
**Algorithm 3** Message-Passing($I_i, N_i$)

---
**Input:** $I_i$ is the message, $N_i$ are the neighbors.
            &bull; Let $R_i$ denote the information received. Initialize $R_i = \{I_i\}$, and send $I_i$ to $N_i$.
            &bull; While $R_i \neq \{I_j, 1 \leq j \leq n\}$:
                If receive message $I_j \notin R_i$, then let $R_i = R_i \cup \{I_j\}$ and send $I_j$ to $N_i$.

---

Lemma 1 bounds (A) by $O(\epsilon)\mathrm{cost}(P, \mathbf{x})$, but we need an accuracy of $\epsilon^2$ to compensate for the $1/\epsilon$ factor in the upper bound of $f_{\mathbf{x}}(p)$. This leads to an $O(1/\epsilon^4)$ factor in the sample complexity.

For (B) and (C), $|\mathrm{cost}(p, \mathbf{x}) - \mathrm{cost}(b_p, \mathbf{x})| > \Delta_p$ since $p \notin G(\mathbf{x})$. This can be used to show that $p$ and $b_p$ are close to each other and far away from $\mathbf{x}$, and thus $|\mathrm{cost}(p, \mathbf{x}) - \mathrm{cost}(b_p, \mathbf{x})|$ is $O(\epsilon)$ smaller than $\mathrm{cost}(p, \mathbf{x})$ and $\mathrm{cost}(b_p, \mathbf{x})$. This fact bounds ((B)) by $O(\epsilon)\mathrm{cost}(P, \mathbf{x})$. It also bounds (C), noting that $\mathbf{E}[\sum_{q \in P_b \cap S} w_q] = |P_b|$, and thus $\sum_{q \in P_b \cap S} w_q \leq 2|P_b|$ when $t \geq O(nk \log \frac{nk}{\delta})$. The proof is completed by bounding the function space dimension by $O(kd)$ as in [8].

# 4 Effect of Network Topology on Communication Cost

If there is a central coordinator in the communication graph, then we can run distributed coreset construction algorithm and send the local portions of the coreset to the coordinator, which can perform the clustering task. The total communication cost is just the size of the coreset.

In this section, we consider distributed clustering over arbitrary connected topologies. We propose to use a message passing approach for collecting information for coreset construction and sharing the local portions of the coreset. The details are presented in Algorithm 2 and 3. Since each piece of the coreset is shared at most twice across any particular edge in message passing, we have

**Theorem 2.** *Given an $\alpha$-approximation algorithm for weighted $k$-means ($k$-median respectively) as a subroutine, there exists an algorithm that with probability at least $1 - \delta$ outputs a $(1 + \epsilon)\alpha$-approximation solution for distributed $k$-means ($k$-median respectively). The communication cost is $O(m(\frac{1}{\epsilon^4}(kd + \log \frac{1}{\delta}) + nk \log \frac{nk}{\delta}))$ for $k$-means, and $O(m(\frac{1}{\epsilon^2}(kd + \log \frac{1}{\delta}) + nk))$ for $k$-median.*

In contrast, an approach where each node constructs an $\epsilon$-coreset for $k$-means and sends it to the other nodes incurs communication cost of $\tilde{O}(\frac{mnkd}{\epsilon^4})$. Our algorithm significantly reduces this.

Our algorithm can also be applied on a rooted tree: we can send the coreset portions to the root which then applies an approximation algorithm. Since each portion are transmitted at most $h$ times,

**Theorem 3.** *Given an $\alpha$-approximation algorithm for weighted $k$-means ($k$-median respectively) as a subroutine, there exists an algorithm that with probability at least $1 - \delta$ outputs a $(1 + \epsilon)\alpha$-approximation solution for distributed $k$-means ($k$-median respectively) clustering on a rooted tree of height $h$. The total communication cost is $O(h(\frac{1}{\epsilon^4}(kd + \log \frac{1}{\delta}) + nk \log \frac{nk}{\delta}))$ for $k$-means, and $O(h(\frac{1}{\epsilon^2}(kd + \log \frac{1}{\delta}) + nk))$ for $k$-median.*

Our approach improves the cost of $\tilde{O}(\frac{nh^4kd}{\epsilon^4})$ for $k$-means and the cost of $\tilde{O}(\frac{nh^2kd}{\epsilon^2})$ for $k$-median in [23] [2]. The algorithm in [23] builds on each node a coreset for the union of coresets from its

children, and thus needs $O(\epsilon/h)$ accuracy to prevent the accumulation of errors. Since the coreset construction subroutine has quadratic dependence on $1/\epsilon$ for $k$-median (quartic for $k$-means), the algorithm then has quadratic dependence on $h$ (quartic for $k$-means). Our algorithm does not build coreset on top of coresets, resulting in a better dependence on the height of the tree $h$.

In a general graph, any rooted tree will have its height $h$ at least as large as half the diameter. For sensors in a grid network, this implies $h = \Omega(\sqrt{n})$. In this case, our algorithm gains a significant improvement over existing algorithms.

## 5 Experiments

Here we evaluate the effectiveness of our algorithm and compare it to other distributed coreset algorithms. We present the $k$-means cost of the solution by our algorithm with varying communication cost, and compare to those of other algorithms when they use the same amount of communication.

**Data sets:** We present results on YearPredictionMSD (515345 points in $\mathbf{R}^{90}, k = 50$). Similar results are observed on five other datasets, which are presented in the supplementary material.

**Experimental Methodology:** We first generate a communication graph connecting local sites, and then partition the data into local data sets. The algorithms were evaluated on Erdös-Renyi random graphs with $p = 0.3$, grid graphs, and graphs generated by the preferential attachment mechanism [1]. We used 100 sites for YearPredictionMSD.

The data is then distributed over the local sites. There are four partition methods: uniform, similarity-based, weighted, and degree-based. In all methods, each example is distributed to the local sites with probability proportional to the site's weight. In uniform partition, the sites have equal weights; in similarity-based partition, each site has an associated data point randomly selected from the global data and the weight is the similarity to the associated point; in weighted partition, the weights are chosen from $|N(0,1)|$; in degree-based, the weights are the sites' degrees.

To measure the quality of the coreset generated, we run Lloyd's algorithm on the coreset and the global data respectively to get two solutions, and compute the ratio between the costs of the two solutions over the global data. The average ratio over 30 runs is then reported. We compare our algorithm with COMBINE, the method of combining coresets from local data sets, and with the algorithm of [23] (Zhang et al.). When running the algorithm of Zhang et al., we restrict the network to a spanning tree by picking a root uniformly at random and performing a breadth first search.

**Results:** Figure 2 shows the results over different network topologies and partition methods. We observe that the algorithms perform well with much smaller coreset sizes than predicted by the theoretical bounds. For example, to get $1.1$ cost ratio, the coreset size and thus the communication needed is only $0.1\% - 1\%$ of the theoretical bound.

In the uniform partition, our algorithm performs nearly the same as COMBINE. This is not surprising since our algorithm reduces to the COMBINE algorithm when each local site has the same cost and the two algorithms use the same amount of communication. In this case, since in our algorithm the sizes of the local samples are proportional to the costs of the local solutions, it samples the same number of points from each local data set. This is equivalent to the COMBINE algorithm with the same amount of communication. In the similarity-based partition, similar results are observed as it also leads to balanced local costs. However, when the local sites have significantly different costs (as in the weighted and degree-based partitions), our algorithm outperforms COMBINE. As observed in Figure 2, the costs of our solutions consistently improve over those of COMBINE by $2\% - 5\%$. Our algorithm then saves $10\% - 20\%$ communication cost to achieve the same approximation ratio.

Figure 3 shows the results over the spanning trees of the graphs. Our algorithm performs much better than the algorithm of Zhang et al., achieving about $20\%$ improvement in cost. This is due to the fact that their algorithm needs larger coresets to prevent the accumulation of errors when constructing coresets from component coresets, and thus needs higher communication cost to achieve the same approximation ratio.

**Acknowledgements**  This work was supported by ONR grant N00014-09-1-0751, AFOSR grant FA9550-09-1-0538, and by a Google Research Award. We thank Le Song for generously allowing us to use his computer cluster.

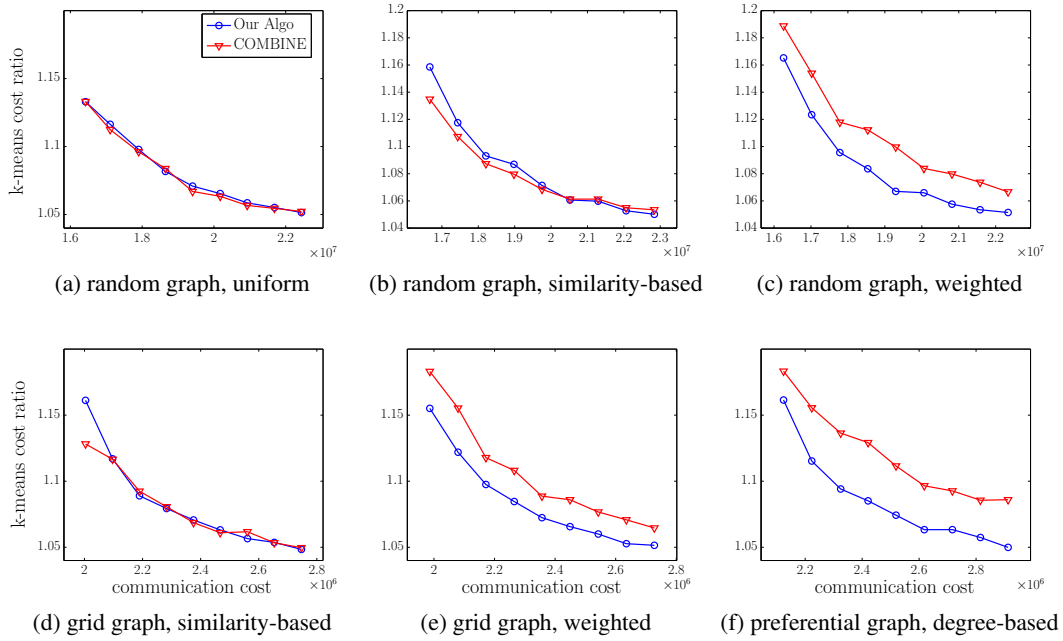

Figure 2: $k$-means cost (normalized by baseline) v.s. communication cost over graphs. The titles indicate the network topology and partition method.

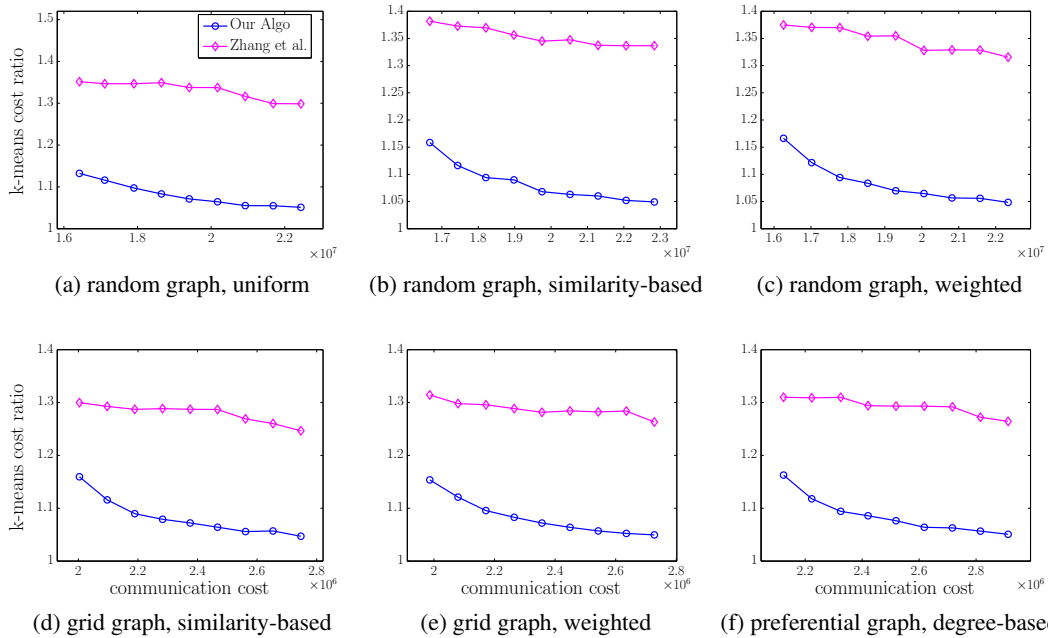

Figure 3: $k$-means cost (normalized by baseline) v.s. communication cost over the spanning trees of the graphs. The titles indicate the network topology and partition method.

## Footnotes

[1]The value that is communicated is the sum of the costs of approximations to the local optimal clustering. This is guaranteed to be no more than a constant factor times larger than the optimal cost.

[2] Their algorithm used coreset construction as a subroutine. The construction algorithm they used builds coreset of size $\tilde{O}(\frac{nkh}{\epsilon^d} \log |P|)$. Throughout this paper, when we compare to [23] we assume they use the coreset construction technique of [8] to reduce their coreset size and communication cost.

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
