[Supplementary Material]

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

## A  Additional Related Work

Many empirical algorithms adapt the centralized algorithms to the distributed setting. They generally provide no bound for the clustering quality or the communication cost. For instance, a technique is proposed in [12] to adapt several iterative center-based data clustering algorithms including Lloyd's algorithm for $k$-means to the distributed setting, where sufficient statistics instead of the raw data are sent to a central coordinator. This approach involves transferring data back and forth in each iteration, and thus the communication cost depends on the number of iterations. Similarly, the communication costs of the distributed clustering algorithms proposed in [9] and [25] depend on the number of iterations. Some other algorithms gather local summaries and then perform global clustering on the summaries. The distributed density-based clustering algorithm in [17] clusters and computes summaries for the local data at each node, and sends the local summaries to a central node where the global clustering is carried out. This algorithm only considers the flat two-tier topology. Some in-network aggregation schemes for computing statistics over distributed data are useful for such distributed clustering algorithms. For example, an algorithm is provided in [7] for approximate duplicate-sensitive aggregates across distributed data sets, such as SUM. An algorithm is proposed in [14] for power-preserving computation of order statistics such as quantile.

Several coreset construction algorithms have been proposed for $k$-median, $k$-means and $k$-line median clustering [16, 6, 15, 20, 10]. For example, the algorithm in [10] constructs a coreset of size $\tilde{O}(kd/\epsilon^2)$ whose cost approximates that of the original data up to accuracy $\epsilon$ with respect to $k$-median in $\mathbf{R}^d$. All of these algorithms consider coreset construction in the centralized setting, while our construction algorithm is for the distributed setting.

There has also been work attempting to parallelize clustering algorithms. [11] showed that coresets could be constructed in parallel and then merged together. In Scalable `k-means++` [3], Bahmani et al. adapted `k-means++` to the parallel setting. In this setting a centralized problem is broken up and distributed to several processors with the aim of reducing computation time. In contrast to the distributed setting, the communication costs are ignored.

There is also related work providing approximation solutions for $k$-median based on random sampling [5]. Particularly, they showed that given a sample of size $\tilde{O}(\frac{k}{\epsilon^2})$ drawn i.i.d. from the data, there exists an algorithm that outputs a solution with an average cost bounded by twice the optimal average cost plus an error bound $\epsilon$. If we convert it to a multiplicative approximation factor, the factor depends on the optimal average cost. When there are outlier points far away from all other points, the optimal average cost can be very small after normalization, then the multiplicative approximation factor is large. The coreset approach provides better guarantees. Additionally, their approach is not applicable to $k$-means.

## B  Proof of Lemma 1

The proof of Lemma 1 follows from the analysis in [10], although not explicitly stated there. We begin with the following theorem for uniform sampling on a function space. The theorem is from [10] but rephrased for convenience.

**Theorem 4** (Theorem 6.9 in [10]). *Let $F$ be a set of functions from $P$ to $\mathbf{R}_{\geq 0}$, and let $\epsilon \in (0, 1)$. Let $S$ be a sample of*

$$|S| = \frac{c}{\epsilon^2}(\dim(F, P) + \log\frac{1}{\delta})$$

*i.i.d items from $P$, where $c$ is a sufficiently large constant. Then, with probability at least $1 - \delta$, for any $f \in F$ and any $r \geq 0$,*

$$\left| \frac{\sum_{p \in P, f(p) \leq r} f(p)}{|P|} - \frac{\sum_{q \in S, f(q) \leq r} f(q)}{|S|} \right| \leq \epsilon r.$$

**Lemma 1.** *Fix a set $F$ of functions $f : P \rightarrow \mathbf{R}_{\geq 0}$. Let $S$ be a sample drawn i.i.d. from $P$ according to $\{m_p \in \mathbf{R}_{\geq 0} : p \in P\}$: for each $q \in S$ and $p \in P$, $q = p$ with probability $\frac{m_p}{\sum_{z \in P} m_z}$. Let $w_p = \frac{\sum_{z \in P} m_z}{m_p |S|}$ for $p \in P$. If for a sufficiently large $c$, $|S| \geq \frac{c}{\epsilon^2}\left(\dim(F, P) + \log\frac{1}{\delta}\right)$, then with probability at least $1 - \delta, \forall f \in F : \left| \sum_{p \in P} f(p) - \sum_{q \in S} w_q f(q) \right| \leq \epsilon \left( \sum_{p \in P} m_p \right) \left( \max_{p \in P} \frac{f(p)}{m_p} \right).$*

*Proof.* Without loss of generality, assume $m_p \in \mathbf{N}^+$. Define $G$ as follows: for each $p \in P$, include $m_p$ copies $\{p_i\}_{i=1}^{m_p}$ of $p$ in $G$ and define $f(p_i) = f(p)/m_p$. Then $S$ is equivalent to a sample draw i.i.d. and uniformly at random from $G$. We now apply Theorem 4 on $G$ and $r = \max_{f \in F, p' \in G} f(p')$. By Theorem 4, we know that for any $f \in F$,

$$\left| \frac{\sum_{p' \in G} f(p')}{|G|} - \frac{\sum_{q' \in S} f(q')}{|S|} \right| \leq \epsilon \max_{p' \in G} f(p'). \tag{1}$$

The lemma then follows from multiplying both sides of (1) by $|G| = \sum_{p \in P} m_p$. Also note that the dimension $\dim(F, G)$ is the same as that of $\dim(F, P)$ as pointed out by [10]. $\square$

## C  Proof of Lemma 2

We have the following lemma for $k$-median with $F = \{f_\mathbf{x} : f_\mathbf{x}(p) = d(p, \mathbf{x}) - d(b_p, \mathbf{x}) + d(p, b_p), \mathbf{x} \in (\mathbf{R}^d)^k\}$.

**Lemma 2.** *For $k$-median, the output of Algorithm 1 is an $\epsilon$-coreset with probability at least $1 - \delta$, if $t \geq \frac{c}{\epsilon^2} \left( \dim(F, P) + \log \frac{1}{\delta} \right)$ for a sufficiently large constant $c$.*

*Proof.* We want to show that for any set of centers $\mathbf{x}$ the true cost for using these centers is well approximated by the cost on the weighted coreset. Note that our coreset has two types of points: sampled points $p \in S = \cup_{i=1}^n S_i$ with weight $w_p := \frac{\sum_{z \in P} m_z}{m_p |S|}$ and local solution centers $b \in B = \cup_{i=1}^n B_i$ with weight $w_b := |P_b| - \sum_{p \in S \cap P_b} w_p$. We use $b_p$ to represent the nearest center to $p$ in the local approximation solution. We use $P_b$ to represent the set of points having $b$ as their closest center in the local approximation solution.

As mentioned above, we construct $f_\mathbf{x}$ to be the difference between the cost of $p$ and the cost of $b_p$ on $\mathbf{x}$ so that Lemma 1 can be applied to $f_\mathbf{x}$. Note that $0 \leq f_\mathbf{x}(p) \leq 2d(p, b_p)$ by triangle inequality, and $S$ is sufficiently large and chosen according to weights $m_p = d(p, b_p)$, so the conditions of Lemma 1 are met. Then we have

$$D = \left| \sum_{p \in P} f_\mathbf{x}(p) - \sum_{q \in S} w_q f_\mathbf{x}(q) \right| \leq 2\epsilon \sum_{p \in P} m_p = 2\epsilon \sum_{p \in P} d(p, b_p) = 2\epsilon \sum_{i=1}^n d(P_i, B_i) \leq O(\epsilon) \sum_{p \in P} d(p, \mathbf{x})$$

where the last inequality follows from the fact that $B_i$ is a constant approximation solution for $P_i$.

Next, we show that the coreset is constructed such that $D$ is exactly the difference between the true cost and the weighted cost of the coreset, which then leads to the lemma.

Note that the centers are weighted such that

$$\sum_{b \in B} w_b d(b, \mathbf{x}) = \sum_{b \in B} |P_b| d(b, \mathbf{x}) - \sum_{b \in B} \sum_{q \in S \cap P_b} w_q d(b, \mathbf{x}) = \sum_{p \in P} d(b_p, \mathbf{x}) - \sum_{q \in S} w_q d(b_q, \mathbf{x}). \tag{2}$$

Also note that $\sum_{p \in P} m_p = \sum_{q \in S} w_q m_q$, so

$$D = \left| \sum_{p \in P} [d(p, \mathbf{x}) - d(b_p, \mathbf{x}) + m_p] - \sum_{q \in S} w_q [d(q, \mathbf{x}) - d(b_q, \mathbf{x}) + m_q] \right|$$

$$= \left| \sum_{p \in P} d(p, \mathbf{x}) - \sum_{q \in S} w_q d(q, \mathbf{x}) - \left[ \sum_{p \in P} d(b_p, \mathbf{x}) - \sum_{q \in S} w_q d(b_q, \mathbf{x}) \right] \right|. \tag{3}$$

By plugging (2) into (3), we have

$$D = \left| \sum_{p \in P} d(p, \mathbf{x}) - \sum_{q \in S} w_q d(q, \mathbf{x}) - \sum_{b \in B} w_b d(b, \mathbf{x}) \right| = \left| \sum_{p \in P} d(p, \mathbf{x}) - \sum_{q \in S \cup B} w_q d(q, \mathbf{x}) \right|$$

which implies the lemma. $\square$

# D  Proof of Theorem 1

**Theorem 1.** *For distributed $k$-means and $k$-median clustering on a graph, there exists an algorithm such that with probability at least $1 - \delta$, the union of its output on all nodes is an $\epsilon$-coreset for $P = \bigcup_{i=1}^{n} P_i$. The size of the coreset is $O(\frac{1}{\epsilon^4}(kd + \log \frac{1}{\delta}) + nk \log \frac{nk}{\delta})$ for $k$-means, and $O(\frac{1}{\epsilon^2}(kd + \log \frac{1}{\delta}) + nk)$ for $k$-median. The total communication cost is $O(mn)$.*

For $k$-median, the statement follows from Lemma 2 and the fact that $\dim(F, P) = O(kd)$ (Theorem 4.8 in [10]).

For $k$-means, we have a similar lemma that when $t = O(\frac{1}{\epsilon^4}(kd + \log \frac{1}{\delta}) + nk \log \frac{nk}{\delta}))$, the algorithm constructs an $\epsilon$-coreset with probability at least $1 - \delta$. The key idea is the same as that for $k$-median: we use centers $b_p$ from the local approximation solutions as an approximation to the original data points $p$, and show that the error between the total cost and the weighted sample cost is approximately the error between the cost of $p$ and its sampled cost (compensated by the weighted centers), which is shown to be small by Lemma 1.

The key difference between $k$-means and $k$-median is that triangle inequality applies directly to the $k$-median cost. In particular, for the $k$-median problem note that $\mathrm{cost}(b_p, p) = d(b_p, p)$ is an upper bound for the error of $b_p$ on any set of centers, i.e. $\forall \mathbf{x} \in (\mathbf{R}^d)^k$, $d(b_p, p) \geq |d(p, \mathbf{x}) - d(b_p, \mathbf{x})| = |\mathrm{cost}(p, \mathbf{x}) - \mathrm{cost}(b_p, \mathbf{x})|$ by triangle inequality. Then we can construct $f_{\mathbf{x}}(p) := \mathrm{cost}(p, \mathbf{x}) - \mathrm{cost}(b_p, \mathbf{x}) + d(b_p, p)$ such that $h_p(\mathbf{x})$ is bounded. In contrast, for $k$-means, the error $|\mathrm{cost}(p, \mathbf{x}) - \mathrm{cost}(b_p, \mathbf{x})| = |d(p, \mathbf{x})^2 - d(b_p, \mathbf{x})^2|$ does not have such an upper bound. The main change to the analysis is that we divide the points into two categories: good points whose costs approximately satisfy the triangle inequality (up to a factor of $1/\epsilon$) and bad points. The good points for a fixed set of centers $\mathbf{x}$ are defined as

$$G(\mathbf{x}) = \{p \in P : |\mathrm{cost}(p, \mathbf{x}) - \mathrm{cost}(b_p, \mathbf{x})| \leq \Delta_p\}$$

where the upper bound is $\Delta_p = \frac{\mathrm{cost}(p, b_p)}{\epsilon}$. Good points we can bound as before. For bad points we can show that while the difference in cost may be larger than $\mathrm{cost}(p, b_p)/\epsilon$, it must still be small, namely $O(\epsilon \min\{\mathrm{cost}(p, \mathbf{x}), \mathrm{cost}(b_p, \mathbf{x})\})$.

Formally, the functions $f_{\mathbf{x}}(p)$ are restricted to be defined only over good points:

$$f_{\mathbf{x}}(p) = \begin{cases} \mathrm{cost}(p, \mathbf{x}) - \mathrm{cost}(b_p, \mathbf{x}) + \Delta_p & \text{if } p \in G(\mathbf{x}), \\ 0 & \text{otherwise.} \end{cases}$$

Then $\sum_{p \in P} \mathrm{cost}(p, \mathbf{x}) - \sum_{q \in S \cup B} w_q \mathrm{cost}(q, \mathbf{x})$ is decomposed into three terms:

$$\sum_{p \in P} f_{\mathbf{x}}(p) - \sum_{q \in S} w_q f_{\mathbf{x}}(q) \tag{4}$$

$$+ \sum_{p \in P \setminus G(\mathbf{x})} [\mathrm{cost}(p, \mathbf{x}) - \mathrm{cost}(b_p, \mathbf{x}) + \Delta_p] \tag{5}$$

$$- \sum_{q \in S \setminus G(\mathbf{x})} w_q [\mathrm{cost}(q, \mathbf{x}) - \mathrm{cost}(b_q, \mathbf{x}) + \Delta_q] \tag{6}$$

Lemma 1 bounds (4) by $O(\epsilon)\mathrm{cost}(P, \mathbf{x})$, but we need an accuracy of $\epsilon^2$ to compensate for the $1/\epsilon$ factor in the upper bound, resulting in a $O(1/\epsilon^4)$ factor in the sample complexity.

We begin by bounding (5). Note that for each term in (5), $|\mathrm{cost}(p, \mathbf{x}) - \mathrm{cost}(b_p, \mathbf{x})| > \Delta_p$ since $p \notin G(\mathbf{x})$. Furthermore, $p \notin G(\mathbf{x})$ only when $p$ and $b_p$ are close to each other and far away from $\mathbf{x}$. In Lemma 3 we use this to show that $|\mathrm{cost}(p, \mathbf{x}) - \mathrm{cost}(b_p, \mathbf{x})| \leq O(\epsilon) \min\{\mathrm{cost}(p, \mathbf{x}), \mathrm{cost}(b_p, \mathbf{x})\}$. The details are presented in the appendix.

Using Lemma 3, (5) can be bounded by $O(\epsilon) \sum_{p \in P \setminus G(\mathbf{x})} \mathrm{cost}(p, \mathbf{x}) \leq O(\epsilon)\mathrm{cost}(P, \mathbf{x})$.

Similarly, by the definition of $\Delta_q$ and Lemma 3, (6) is bounded by

$$
(6) \quad \leq \quad \sum_{q \in S \backslash G(\mathbf{x})} 2w_q |\mathrm{cost}(q, \mathbf{x}) - \mathrm{cost}(b_q, \mathbf{x})| \leq O(\epsilon) \sum_{q \in S \backslash G(\mathbf{x})} w_q \, \mathrm{cost}(b_q, \mathbf{x})
$$

$$
\leq \quad O(\epsilon) \sum_{b \in B} \left( \sum_{q \in P_b \cap S} w_q \right) \mathrm{cost}(b, \mathbf{x}).
$$

Note that the expectation of $\sum_{q \in P_b \cap S} w_q$ is $|P_b|$. By a sampling argument (Lemma 4), if $t \geq O(nk \log \frac{nk}{\delta})$, then $\sum_{q \in P_b \cap S} w_q \leq 2|P_b|$. Then (6) is bounded by $O(\epsilon) \sum_{b \in B} \mathrm{cost}(b, \mathbf{x})|P_b| = O(\epsilon) \sum_{p \in P} \mathrm{cost}(b_p, \mathbf{x})$ where $\sum_{p \in P} \mathrm{cost}(b_p, \mathbf{x})$ is at most a constant factor more than the optimum cost.

Since each of (4),(5), and (6) is $O(\epsilon)\mathrm{cost}(P, \mathbf{x})$, we know that their sum is the same magnitude. Combining the above bounds, we have $\left| \mathrm{cost}(P, \mathbf{x}) - \sum_{q \in S \cup B} w_q \mathrm{cost}(q, \mathbf{x}) \right| \leq O(\epsilon)\mathrm{cost}(P, \mathbf{x})$. The proof is then completed by choosing a suitable $\epsilon$, and bounding $\dim(F, P) = O(kd)$ as in [10].

**Lemma 3.** *If $d(p, b_p)^2/\epsilon \leq |d(p, \mathbf{x})^2 - d(b_p, \mathbf{x})^2|$, then*

$$
|d(p, \mathbf{x})^2 - d(b_p, \mathbf{x})^2| \leq 8\epsilon \min\{d(p, \mathbf{x})^2, d(b_p, \mathbf{x})^2\}.
$$

*Proof.* We first have by triangle inequality

$$
|d(p, \mathbf{x})^2 - d(b_p, \mathbf{x})^2| \leq d(p, b_p)[d(p, \mathbf{x}) + d(b_p, \mathbf{x})].
$$

Then by $d(p, b_p)^2/\epsilon \leq |d(p, \mathbf{x})^2 - d(b_p, \mathbf{x})^2|$,

$$
d(p, b_p) \leq \epsilon[d(p, \mathbf{x}) + d(b_p, \mathbf{x})].
$$

Therefore, we have

$$
\begin{aligned}
|d(p, \mathbf{x})^2 - d(b_p, \mathbf{x})^2| &\leq d(p, b_p)[d(p, \mathbf{x}) + d(b_p, \mathbf{x})] \leq \epsilon[d(p, \mathbf{x}) + d(b_p, \mathbf{x})]^2 \\
&\leq 2\epsilon[d(p, \mathbf{x})^2 + d(b_p, \mathbf{x})^2] \leq 2\epsilon[d(p, \mathbf{x})^2 + (d(p, \mathbf{x}) + d(p, b_p))^2] \\
&\leq 2\epsilon[d(p, \mathbf{x})^2 + 2d(p, \mathbf{x})^2 + 2d(p, b_p)^2] \leq 6\epsilon d(p, \mathbf{x})^2 + 4\epsilon d(p, b_p)^2 \\
&\leq 6\epsilon d(p, \mathbf{x})^2 + 4\epsilon^2 |d(p, \mathbf{x})^2 - d(b_p, \mathbf{x})^2|
\end{aligned}
$$

for sufficiently small $\epsilon$. Then

$$
|d(p, \mathbf{x})^2 - d(b_p, \mathbf{x})^2| \quad \leq \quad \frac{6\epsilon}{1 - 4\epsilon^2} d(p, \mathbf{x})^2 \leq 8\epsilon d(p, \mathbf{x})^2.
$$

Similarly, $|d(p, \mathbf{x})^2 - d(b_p, \mathbf{x})^2| \leq 8\epsilon d(b_p, \mathbf{x})^2$. The lemma follows from the last two inequalities. $\square$

**Lemma 4** (Corollary 15.4 in [10]). *Let $0 < \delta < 1/2$, and $t \geq c|B| \log \frac{|B|}{\delta}$ for a sufficiently large $c$. Then with probability at least $1 - \delta$, $\forall b \in B_i$, $\sum_{q \in P_b \cap S} w_q \leq 2|P_b|$.*

# E    Complete Experimental Results

Here we present the results of all the data sets over different network topologies and data partition methods.

Figure 4 shows the results of all the data sets on random graphs. The first column of Figure 4 shows that our algorithm and COMBINE perform nearly the same in the uniform data partition. This is not surprising since our algorithm reduces to the COMBINE algorithm when each local site has the same cost and the two algorithms use the same amount of communication. In this case, since in our algorithm the sizes of the local samples are proportional to the costs of the local solutions, it samples the same number of points from each local data set. This is equivalent to the COMBINE algorithm with the same amount of communication. In the similarity-based partition, similar results are observed as this partition method also leads to balanced local costs. However, in the weighted partition where local sites have significantly different contributions to the total cost, our algorithm

outperforms COMBINE. It improves the $k$-means cost by $2\% - 5\%$, and thus saves $10\% - 30\%$ communication cost to achieve the same approximation ratio.

Figure 5 shows the results of all the data sets on grid and preferential graphs. Similar to the results on random graphs, our algorithm performs nearly the same as COMBINE in the similarity-based partition and outperforms COMBINE in the weighted partition and degree-based partition. Furthermore, Figure 4 and 5 also show that the performance of our algorithm merely changes over different network topologies and partition methods.

Figure 6 shows the results of all the data sets on the spanning trees of the random graphs and Figure 7 shows those on the spanning trees of the grid and preferential graphs. Compared to the algorithm of Zhang et al., our algorithm consistently shows much better performance on all the data sets in different settings. It improves the $k$-means cost by $10\% - 30\%$, and thus can achieve even better approximation ratio with only $10\%$ communication cost. This is because the algorithm of Zhang et al. constructs coresets from component coresets and needs larger coresets to prevent the accumulation of errors. Figure 6 also shows that although their costs decrease with the increase of the communication, the decrease is slower on larger graphs (e.g., as in the experiments for YearPredictionMSD). This is due to the fact that the spanning tree of a larger graph has larger height, leading to more accumulation of errors. In this case, more communication is needed to prevent the accumulation.

Figure 4: $k$-means cost on random graphs. Columns: random graph with uniform partition, random graph with similarity-based partition, and random graph with weighted partition. Rows: Spam, Pendigits, Letter, synthetic, ColorHistogram, and YearPredictionMSD.

Figure 5: $k$-means cost on grid and preferential graphs. Columns: grid graph with similarity-based partition, grid graph with weighted partition, and preferential graph with degree-based partition. Rows: Spam, Pendigits, Letter, synthetic, ColorHistogram, and YearPredictionMSD.

Figure 6: $k$-means cost on the spanning trees of the random graphs. Columns: random graph with uniform partition, random graph with similarity-based partition, and random graph with weighted partition. Rows: Spam, Pendigits, Letter, synthetic, ColorHistogram, and YearPredictionMSD.

Figure 7: *k*-means cost on the spanning trees of the grid and preferential graphs. Columns: grid graph with similarity-based partition, grid graph with weighted partition, and preferential graph with degree-based partition. Rows: Spam, Pendigits, Letter, synthetic, ColorHistogram, and YearPredictionMSD.