[Reviews · NeurIPS 2013]

Submitted by Assigned_Reviewer_6

The paper provides provably efficient algorithms for performing k-means and k-median clustering in the distributed setting.

The main focus of the paper is minimizing communication cost in the distributed network. Although, i am not very much aware of the literature, the paper seems to provide a very novel idea of distributed coresets that leads to clustering algorithms which provably improves the state of the art communication complexity significantly.

Existing approaches only use the idea of approximating coresets by taking the union of local coresets. The key idea in this papers is a construction of distributed coresets which is different from taking the union and this can be of independent interest in itself. The theoretical ideas in are based on the notion of dimension of the function space and the sampling lemma.

The paper also provides a rigorous experimental evaluation and shows that the proposed algorithms outperforms existing state of art methods in terms of communication complexity supporting the theory.

One concern is that in experiments the accuracy of the clustering algorithms is not compared (say for the same communication budget).
Summary: A good paper about minimizing communication cost of distributed clustering, with solid theoretical & experimental support.

Submitted by Assigned_Reviewer_7

This paper presents new distributed algoritms for clustering methods that use centers as cluster representation. The new algorithms are shown to reduce the communication overhead. At the same communication cost, the new methods improve clustering objectives for both synthetic and three UCI datasets.

The paper is very hard-core. More explanation about the derivation clue and the values of the theorems would be helpful.

I am a little bit skeptical on the approximation quality of coreset when the dimensionality increases. Datasets in the current experiments are most low-dimensional.

The title is too large. The work is actually on some particular clustering methods that uses centers and coresets. The word "graphs" is also vague, which does not reveal any highlights or distinguishable points.
Summary: Improved methods for center-based distributed clustering are presented. The paper is very hard-core.

Submitted by Assigned_Reviewer_8

This paper proposes new distributed algorithms for k-means and k-median methods. Its main idea is to construct a global coreset in a distributed fashion using cost information from local approximations. This paper is well written with solid content. It proposes distributed protocol for k-means/median clustering based on coreset construction, provides theoretical guarantee on the approximation and upper bounds on the communication cost, performs a couple of experiments to show the proposed approach outperforms the existing ones.

Having said, the paper also comes with several caveats. First of all, it is not very clear what is the main technical contribution of the paper, especially given the existing works in [5], [10] and [19]. Using coreset idea for approximating clustering algorithms is well studied (e.g., in [5],[10]), and a distributed clustering algorithm based on coreset is proposed in [19]. I clearly see that this paper contributes a distributed method for constructing coreset independent of the network topology, and develops a better algorithm than that in [19], by reducing the communication cost by a factor of sqrt(n). However, the distributed coreset construction in Algorithm 1 and the related bounds seems to be a straightforward application of results in [5].

It is nice to see the theoretical analysis on the algorithm and the upper bounds on coreset size and communication cost. However, the upper bounds seem to be too loose to use in practical applications. Given a dataset and its location over a distributed network, how could you determine the size of coreset? The bound for achieving this goal for k-means is along the line of O(kd/(e^4)), which could be huge even for moderate large k, d and relative small e. For example, for k=10, d = 100, e = 0.1, the size of coreset predicted by your bound is 10 million, which could be way more larger than the actual data size.

In the experiments, the number of distributed sites is small (e.g., 10 for Pendigits, 25 for ColorHistogram). Have you tried experiments with larger number of distributed sites? In addition, the three datasets used are relative-low dimensional. Would this kind of approach work for high-dimensional sparse data?
Summary: This paper is well-written with solid content, but also comes with several caveats, including unclear technical contribution and loose upper bounds.
Author Feedback

Author rebuttal: ----------------------------------------
Review 1 Assigned_Reviewer_6
----------------------------------------

Thank you for your insightful comments.

We would like to clarify a point concerning your question:

"One concern is that in experiments the accuracy of the clustering algorithms is not compared (say for the same communication budget). "

In this work, we focus on the classic k-median/k-means objectives and our theoretical guarantees are all about the cost of these objectives (Theorem 1 and Theorem 2). Therefore, in the experimental section we compare the k-means quality of the solutions obtained by our algorithm and the other two competitors (Zhang's algorithm and COMBINE) given the same communication budget; we of course also vary the communication budget. Please see lines 356-360 and Figures 2 and 3.

We are happy to further clarify this in the final version of the paper.

----------------------------------------
Review 2 Assigned_Reviewer_7
----------------------------------------

Thank you for your comments.

We would like to address your concern regarding the presentation of the paper.

Despite the space limitation, we have spent significant effort in providing both rigorous proofs and guiding intuitions. For example, for understanding the dimension of a function space, which is the key notion in analyzing the coreset construction, we describe in Section 3 (lines 191 -203) its relation with the standard VC-dimension. Also, we present a few paragraphs (lines 212-232) describing the intuition about bounding the error of the sample (and subsequently the coreset). We will be happy to incorporate any further explanations you suggest in the final version of the paper.

We will also make our title more specific, such as "Distributed k-median/k-means clustering on general communication graphs".

For the experimental section, we would like to point out that the size of the coreset has a linear dependence on the dimension. This implies that the approach also works for the high dimensional case.

----------------------------------------
Review 3 Assigned_Reviewer_8
----------------------------------------

Thank you for your detailed comments. We would like to address your comments concerning the "unclear technical contributions and loose upper bounds".

While our results build on tools in [5] (for example Lemma 1), our construction and analysis require novel ideas. One key technical contribution we make is to show that the coreset construction can be performed in a distributed manner while achieving the same bounds on the coreset size as in the centralized setting. In our distributed coreset construction (lines 162-173), the points are sampled according to their contributions to the local solution (as opposed to that of a global solution as in the analysis of [5] in the centralized setting), which makes *low communication* possible. The results in [5] do not imply that such a local sampling scheme leads to the same bounds in the centralized setting; we show that it is indeed the case. For added clarity, we will explicitly point out the novelty of our results in the introduction in the final version of the paper.

Another major contribution is to provide a much better empirical evaluation of the coreset-based approach for distributed clustering compared to the prior work. In particular, in contrast to our thorough evaluation, [5] has no experiments and [19] only has experiments for data with dimension 1.

Additionally, we would like to address your concern about loose upper bounds.

Our theoretical bounds and experimental results are complementary to each other and they can help provide a clear picture for the benefits of our method. The key point of the theoretical bounds is that the coreset size can be independent of the actual data size, and linear in the dimension and the number of clusters. The key point of our experimental results is that our algorithm performs better than predicted by the theoretical bounds, and outperforms other coreset construction algorithms. As we point out in lines 362-364, we do find that in practice our algorithm needs a communication cost less than 1% of the bounds. Furthermore, it improves the costs of the other algorithms and thus saves 10%-30% communication cost to achieve the same accuracy (lines 369-372).

Concerning your question about high dimensional data, we would like to point out that the size of the coreset has a linear dependence on the dimension. This implies that the approach also works for the high dimensional case. Experiments on larger graphs using higher dimensional data will be conducted and the results will be added to the final version of the paper.